# Differences in healthcare expenditures for inflammatory bowel disease by insurance status, income, and clinical care setting

Michelle D. Park[1], Jay Bhattacharya[2] and KT Park[3]

[1] School of Medicine, Stanford University, Stanford, CA, USA
[2] Department of Medicine, Department of Economics, Center for Health Policy/Primary Care Outcomes Research, Stanford University, Stanford, CA, USA
[3] Division of Pediatric Gastroenterology, Department of Pediatrics, Center for Health Policy/Primary Care Outcomes Research, Stanford University, Stanford, CA, USA

## ABSTRACT

**Background.** Socioeconomic factors and insurance status have not been correlated with differential use of healthcare services in inflammatory bowel disease (IBD).

**Aim.** To describe IBD-related expenditures based on insurance and household income with the use of inpatient, outpatient, emergency, and office-based services, and prescribed medications in the United States (US).

**Methods.** We evaluated the Medical Expenditure Panel Survey from 1996 to 2011 of individuals with Crohn's disease (CD) or ulcerative colitis (UC). Nationally weighted means, proportions, and multivariate regression models examined the relationships between income and insurance status with expenditures.

**Results.** Annual per capita mean expenditures for CD, UC, and all IBD were \$10,364 ($N = 238$), \$7,827 ($N = 95$), and \$9,528, respectively, significantly higher than non-IBD (\$4,314, $N = 276,372$, $p < 0.05$). Publicly insured patients incurred the highest costs (\$18,067) over privately insured (\$8,014, $p < 0.05$) or uninsured patients (\$5,129, $p < 0.05$). Among all IBD patients, inpatient care composed the highest proportion of costs (\$3,392, $p < 0.05$). Inpatient costs were disproportionately higher for publicly insured patients. Public insurance had higher odds of total costs than private (OR 2.13, CI [1.08–4.19]) or no insurance (OR 4.94, CI [1.26–19.47]), with increased odds for inpatient and emergency care. Private insurance had higher costs associated with outpatient care, office-based care, and prescribed medicines. Low-income patients had lower costs associated with outpatient (OR 0.38, CI [0.15–0.95]) and office-based care (OR 0.21, CI [0.07–0.62]).

**Conclusions.** In the US, high inpatient utilization among publicly insured patients is a previously unrecognized driver of high IBD costs. Bridging this health services gap between SES strata for acute care services may curtail direct IBD-related costs.

Corresponding author
KT Park, ktpark@stanford.edu

## INTRODUCTION

Inflammatory bowel disease (IBD), consisting of Crohn's disease (CD) and ulcerative colitis (UC), is an especially costly chronic disease affecting nearly one million Americans and increasing in prevalence, with disproportionate increases in racial and ethnic minorities (*Molodecky et al., 2012*; *Nguyen et al., 2006*; *Straus et al., 2000*). IBD is a major chronic disease with per-patient yearly expenditures estimated around $8,265–$11,129 for CD, more costly than diabetes, stroke, coronary artery disease, chronic obstructive pulmonary disease, or multiple sclerosis (*Gunnarsson et al., 2012*; *Kappelman et al., 2007*).

IBD care spans a particularly wide range of services from inpatient, outpatient, emergency, and office-based settings, and unequal utilization of necessary services by different patient populations carries the potential to create economic waste, avoidable morbidity, and health disparities (*Sewell & Velayos, 2013*). In addition, increasing use of medical therapeutics for IBD, in particular biologic agents, creates new opportunities for costs to rapidly incur (*Swoger & Binion, 2010*; *Benchimol et al., 2011*).

Race and socioeconomic factors have long been shown to be associated with unequal healthcare access and utilization, with economic and health implications (*Andrulis, 1998*). As supported by existing literature, we noted a trend for nonwhite, poor, and underinsured patients to utilize less outpatient care and more inpatient care. Black patients utilized less ambulatory care, specialists, and biologics than whites, while exhibiting increased hospitalization rates (*Sewell, Yee & Inadomi, 2010*; *Nguyen et al., 2010*; *Jackson et al., 2008*; *Flasar et al., 2008*). Race-related health disparities have also been demonstrated in IBD disease phenotype, surgery rates, type of surgery, perianal fistulizing disease, and extraintestinal manifestations (*Nguyen et al., 2006*; *Basu et al., 2005*). Lower income was associated with higher rates of CD-related surgery along with higher IBD-related hospitalizations, emergency department (ED) visits, and physician visits (*Benchimol et al., 2011*; *Nahon et al., 2009*). When comparing race against socioeconomic factors, insurance status was a stronger predictor of leaving against medical advice than race (*Kaplan et al., 2009*). However, many of these studies failed to separate socioeconomic factors from race/ethnicity, and all were limited in scope by focusing either on a few centers or on one clinical care setting, precluding generalizability and comparisons between different types of services.

Of note, few of the current studies on socioeconomic or racial/ethnic differences in IBD contained nationally representative sample sets. The Medical Expenditure Panel Survey (MEPS) is a nationally representative database that samples 15,000 individuals every year (*Stone, 2012*). It is possibly the most comprehensive dataset on U.S. health services and expenditures, capturing insurer costs as well as out-of-pocket expenses and including many relevant comorbid diseases (*Gunnarsson et al., 2012*; *Stone, 2012*).

We aimed to characterize differences in expenditures based on insurance status, income, and race/ethnicity as they may be associated with differential use of inpatient, outpatient, emergency, and office-based services, as well as prescribed IBD medications. We hypothesized that publicly insured, uninsured, and nonwhite patients would utilize disproportionately more acute care as defined by inpatient and emergency services, while

privately insured and white patients would utilize disproportionately more non-acute care as defined by outpatient and office-based services, and prescribed medicines.

## METHODS

### Data

We performed a longitudinal analysis on data from 1996 to 2011 in the Household Component of MEPS, a nationally representative database conducted by the Agency for Healthcare Research and Quality. MEPS collects data on healthcare utilization and expenditures, health status, health insurance coverage, income, employment, and socio-demographic characteristics for the civilian, non-institutionalized population. 15,000 new individuals are sampled each year and followed for two years with in-person interviews, with response rates ranging from 54 to 78% (*Agency for Healthcare Research and Quality, 2013a*). MEPS utilizes a complex sampling methodology that includes stratification, clustering, multistage selection, and oversampling of certain subgroups including racial/ethnic minorities (*Machlin, Yu & Zodet, 2005*). Survey weights allow for nationally representative data analyses and the weighting process includes adjustments for nonresponse over time along with calibration to independent population figures from the U.S. Census Bureau's Current Population Survey (*Machlin, Yu & Zodet, 2005*; *Agency for Healthcare Research and Quality, 2010*).

MEPS defines inpatient, emergency, and outpatient visits as occurring in a hospital setting or a facility connected with a hospital (*Agency for Healthcare Research and Quality, 2009*). Outpatient visits are defined as not requiring overnight hospitalization, as opposed to inpatient visits. Office-based events do not occur in a hospital or hospital-connected facility, but can occur in a variety of settings including doctor's or group practice office, medical clinic, surgical center, community health center, walk-in urgent care centers, or laboratory/X-ray facilities (*Agency for Healthcare Research and Quality, 2009*). Thus, both outpatient and office-based care may include general primary care, and both may involve same-day procedures.

Self-reported expenditure data are validated with information from healthcare and pharmaceutical providers. Self-reported medical conditions are mapped by professional coders to International Classification of Diseases, Ninth Revision, Clinical Modification (ICD-9-CM) diagnostic codes (*Agency for Healthcare Research and Quality, 2013b*).

### Study population and variables

Individuals ages 3–90 with ICD-9-CM codes of 555.x or 556.x were included in this study a priori and defined as having CD or UC, respectively. Individuals lacking person-level weights were excluded.

Demographic data included age, sex, race/ethnicity, and poverty status. Race/ethnicity was encoded as non-Hispanic white (subsequently abbreviated to "white") or non-white, which included black, Hispanic, Asian, Native American, and mixed-race individuals. Poverty status was measured as a binary variable comparing poor patients to not poor

patients, with poverty defined as having a family income less than 100% of the federal poverty line (FPL) defined by the U.S. Census Bureau's Current Population Survey.

The health-related quality of life comorbidity index (HRQL-CI) was used to adjust for comorbid conditions. The HRQL-CI is a validated risk adjustment index that outperforms the Charlson comorbidity index when external validation was assessed in MEPS (*Mukherjee et al., 2011*; *Ou et al., 2012*). To form the HRQL-CI, *Mukherjee et al. (2011)* selected 44 adult, gender-neutral, chronic conditions, then identified those significantly associated with the Short Form-12 physical component summary and mental component summary. The resulting two subsets of conditions comprise the HRQL-CI, consisting of a physical component score and a mental component score.

Insurance status was measured as a series of binary variables comparing private, public, and no insurance, for individuals who maintained the same insurance category for a full year. The definition of public insurance in MEPS included Medicaid, Medicare, Tricare (U.S. Department of Defense Military Health System), State Children's Health Insurance Program (SCHIP), and other public hospital/physician programs (*Agency for Healthcare Research and Quality, 2013b*). Private insurance was non-public insurance that covered hospital and physician care. Individuals only covered by single-service plans (e.g. drug, dental, or vision plans) were considered uninsured.

IBD-related medications were identified using pharmacy-reported prescription names. We identified immunomodulators—that is, thiopurines (6-mercaptopurine and azathioprine) and methotrexate—anti-tumor necrosis factor (anti-TNF) agents (adalimumab), 5-aminosalicylate agents, prednisone, antibiotics (metronidazole and ciprofloxacin), and other IBD-related medicines (e.g., laxatives, anti-diarrheals, proton pump inhibitors, and histamine H2 receptor antagonists) as identified by gastroenterology-specific clinical judgment.

## Statistical analyses

The primary dependent variables were health expenditures—in total and subcategorized into prescribed medicines or mutually-exclusive clinical care settings (inpatient, outpatient, emergency, and office-based). The primary independent variables were insurance status and poverty status. In calculating standard errors, we accounted for the complex sampling design of MEPS using Stata version 12 (Statacorp, College Station, TX). Sampling variances were estimated using Taylor series linearization (delta method).

Means and proportions were used to produce summary statistics. Multivariate logistic regression models examined the likelihood of incurring annual per capita expenditures above the mean for each respective category (total expenditures, prescribed medicines, or specific clinical care settings). The covariates were age, sex, race/ethnicity, and comorbidities as measured by the HRQL-CI.

**Table 1  Characteristics of IBD patients.**

| | All respondents ($n = 276,702$) | IBD ($n = 333$) | CD ($n = 238$) | UC ($n = 95$) |
|---|---|---|---|---|
| **Treated prevalence (no. per 100,000)** | – | 238 | 165 | 73 |
| **Female (%)** | 55 | 48 | 43 | 59 |
| **Male (%)** | 45 | 52 | 57 | 41 |
| **Age (mean) (s.e.)** | 39.2 (0.2) | 46.2 (1.3) | 47.0 (1.6) | 44.4 (1.8) |
| **Age (%)** | | | | |
| 0–18 | 24 | 4 | 5 | 3 |
| 19–39 | 26 | 31 | 30 | 35 |
| 40–64 | 34 | 51 | 48 | 56 |
| 65+ | 17 | 15 | 18 | 7 |
| **Race/Ethnicity (%)** | | | | |
| Non-hispanic white | 73 | 88 | 90 | 84 |
| Black | 13 | 6 | 6 | 7 |
| Hispanic | 13 | 4 | 2 | 8 |
| **Family income as % of federal poverty line[*] (%)** | | | | |
| Poor (<100%) | 12 | 9 | 11 | 4 |
| Near poor (100% to <125%) | 4 | 3 | 3 | 2 |
| Low income (125% to <200%) | 13 | 13 | 13 | 13 |
| Middle income (200% to <400%) | 31 | 27 | 25 | 30 |
| High income (≥400%) | 39 | 48 | 47 | 51 |
| **Insurance[**] (%)** | | | | |
| Private | 33 | 47 | 42 | 58 |
| Public | 20 | 16 | 20 | 7 |
| Uninsured | 8 | 8 | 9 | 6 |
| **HRQL-CI (mean) (s.e.)** | 1.78 (.01) | 2.06 (.16) | 2.26 (.21) | 1.62 (.20) |

**Notes.**

[*] As defined by the Current Population Survey.

[**] Defined as maintaining the insurance category for a full year. Values are nationally representative except $n$'s.

## RESULTS

### Characteristics of IBD patients

We identified 238 individuals with CD, 95 with UC, and 276,369 individuals without IBD (Table 1). MEPS only collects information on conditions associated with medical events, so treated prevalence for CD was 0.17% when weighted to the U.S. population and 0.07% for UC. Unless noted, all subsequent values also refer to nationally representative estimates. The mean age was 47 for CD and 44 for UC, and 43% of CD patients and 59% of UC patients were female.

Compared to the overall population, patients with IBD were more likely to be white (88% vs. 73%) and less likely to be black and Hispanic (6% and 4% respectively vs. 13% and 13%) (Table 1). Those with IBD were also more likely to be in the highest income bracket of ≥400% FPL (48% vs. 39% of the overall population), and more likely to hold private insurance all year (47% vs. 33% of the overall population). The proportions of IBD patients holding public and no insurance were comparable to the overall population.

**Table 2  Distribution of expenditures across clinical care settings by diagnosis.**

| | IBD (n = 333) | | CD (n = 238) | | UC (n = 95) | |
|---|---|---|---|---|---|---|
| **All Expenditures (OOP + Insurer)** | **Expenditures (mean) (s.e.)** | **% of total** | **Expenditures (mean) (s.e.)** | **% of total** | **Expenditures (mean) (s.e.)** | **% of total** |
| **Total** | 9,528 (910) | – | 10,364 (1,173) | – | 7,827 (1,182) | – |
| **Acute care** | | | | | | |
| Inpatient | 3,392 (578) | 36 | 3,743 (743) | 36 | 2,722 (810) | 35 |
| Emergency | 252 (53) | 3 | 283 (73) | 3 | 192 (47) | 2 |
| **Non-acute care** | | | | | | |
| Outpatient | 1,180 (237) | 12 | 1,166 (253) | 11 | 1,241 (529) | 16 |
| Office-based | 1,705 (163) | 18 | 1,892 (212) | 18 | 1,269 (205) | 16 |
| **Rx medicines** | 711 (106) | 7 | 802 (143) | 8 | 471 (97) | 6 |
| | **IBD (n = 333)** | | **CD (n = 238)** | | **UC (n = 95)** | |
| **OOP expenditures** | **OOP (mean) (s.e.)** | **% of total** | **OOP (mean) (s.e.)** | **% of total** | **OOP (mean) (s.e.)** | **% of total** |
| **Total** | 1,061 (80) | – | 1,088 (78) | – | 982 (187) | – |
| **Acute care** | | | | | | |
| Inpatient | 48 (17) | 5 | 58 (24) | 5 | 29 (14) | 3 |
| Emergency | 29 (9) | 3 | 39 (13) | 4 | 9 (4) | 1 |
| **Non-acute care** | | | | | | |
| Outpatient | 99 (32) | 9 | 75 (18) | 7 | 151 (91) | 15 |
| Office-based | 219 (28) | 21 | 222 (31) | 20 | 182 (36) | 19 |
| **Rx medicines** | 150 (18) | 14 | 169 (24) | 15 | 100 (18) | 10 |

**Notes.**

Means are per capita, per year. OOP, out-of-pocket. Values are nationally representative except *n*'s.

Mean HRQL-CI scores were 2.06 for IBD patients (SE 0.16) and 1.78 for all respondents (SE 0.01).

## Direct cost burden of IBD by clinical care setting

Annual per capita mean expenditures for CD, UC, and all IBD were $10,364, $7,827, and $9,528, respectively, each significantly higher than non-IBD expenditures ($4,314, $p < 0.05$) by $3–6 K more per year (Table 2). Among IBD patients, inpatient mean expenditures ($3,392, SE 578) composed the highest proportion of direct costs, above outpatient, office-based, emergency, or prescribed medicines ($p < 0.05$) and nearly double the next closest subcategory of office-based expenditures ($1,705, SE 163) (Table 2; Fig. 1). In contrast, emergency expenditures ($252, SE 53) composed the lowest proportion of direct costs ($p < 0.05$).

In terms of out-of-pocket (OOP) costs, annual per capita mean expenditures for all IBD were again significantly higher than for non-IBD ($1,061 vs. $597, $p < 0.05$) (Table 2). Although inpatient costs contributed the greatest amount to total IBD expenditures as described above, when considering OOP costs, inpatient (mean $48, SE 17) contributed less than outpatient, office-based, and prescribed medicine costs. The greatest OOP contribution came from office-based (mean $219, SE 28) and prescribed medicine costs

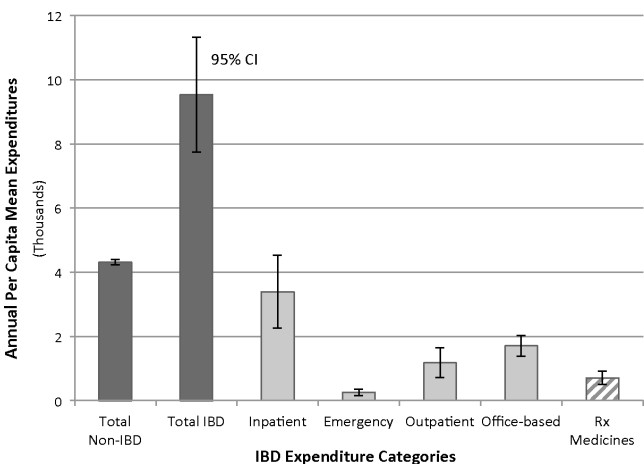

**Figure 1 Distribution of annual per capita mean expenditures across various categories.** IBD patients unless specified non-IBD. Dark gray: total expenditures. Light gray: mutually exclusive clinical care settings. Error bars are 95% confidence intervals.

(mean \$150, SE 18), while emergency costs contributed the least to OOP expenditures (mean \$29, SE 9).

## Direct cost burden of publicly vs. privately insured IBD patients

When examining the effect of insurance status on annual per capita mean expenditures, publicly insured IBD patients had the highest direct costs by over \$10 K (\$18,067), over double that of privately insured (\$8,014, $p < 0.05$) and uninsured patients (\$5,129, $p < 0.05$) (Table 3). For those publicly insured patients, the vast majority of their high expenditures derived from inpatient costs, at 5x or \$7.8 K more than the next closest subcategory of office-based costs (mean \$9,790 vs. \$1,941, $p < 0.05$) (Table 3; Fig. 2A). For privately insured or uninsured patients, however, inpatient costs were not significantly greater than any other subcategories.

When comparing mean expenditures between private and public insurance in each subcategory, only the inpatient subcategory exhibited a significant difference. Publicly insured patients spent 4.5x or \$7.6 K more than the privately insured (mean \$9,790 vs. \$2,174, $p < 0.05$) (Table 3; Fig. 2A). All other clinical settings and prescribed medicine costs were comparable between IBD patients with public and private insurance.

## Effects of no insurance and race/ethnicity on IBD expenditures

Due to the small sample size of uninsured IBD patients, mean expenditures by clinical care setting showed little statistical significance against publicly or privately insured patients (Table 3). Office-based visits, however, showed that the uninsured spent significantly less (mean \$529, SE 152) than either the privately insured (mean \$1801, SE 256, $p < 0.05$) or the publicly insured (mean \$1941, SE 435, $p < 0.05$) by factors of 3.5 and 3.7, respectively (Table 3; Fig. 2A).

**Table 3  Association between insurance status and expenditures across clinical care.**

| | Private (n = 136) | | Public (n = 63) | | Uninsured (n = 26) | |
|---|---|---|---|---|---|---|
| All expenditures (OOP + Insurer) | Expenditures (mean) (s.e.) | % of total | Expenditures (mean) (s.e.) | % of total | Expenditures (mean) (s.e.) | % of Total |
| Total | 8,014 (918) | – | 18,067 (3,918) | – | 5,129 (1,675) | – |
| **Acute care** | | | | | | |
| Inpatient | 2,174 (609) | 27 | 9,790 (2,735) | 54 | 2,840 (1,585) | 55 |
| Emergency | 217 (65) | 3 | 591 (258) | 3 | 235 (73) | 5 |
| **Non-acute care** | | | | | | |
| Outpatient | 1,275 (399) | 16 | 1,696 (917) | 9 | 256 (111) | 5 |
| Office-based | 1,801 (256) | 22 | 1,941 (435) | 11 | 529 (152) | 10 |
| **Rx medicines** | 769 (227) | 3 | 515 (100) | 3 | 430 (194) | 8 |
| | **Private (n = 136)** | | **Public (n = 63)** | | **Uninsured (n = 26)** | |
| OOP expenditures | OOP (mean) (s.e.) | % of total | OOP (mean) (s.e.) | % of total | OOP (mean) (s.e.) | % of total |
| Total | 1,063 (128) | – | 1,157 (180) | – | 1,220 (281) | – |
| **Acute care** | | | | | | |
| Inpatient | 26 (11) | 2 | 38 (15) | 3 | 77 (48) | 6 |
| Emergency | 15 (6) | 1 | 47 (38) | 4 | 121 (58) | 10 |
| Non-acute care | | | | | | |
| Outpatient | 147 (70) | 14 | 49 (21) | 4 | 94 (92) | 8 |
| Office-based | 281 (49) | 26 | 136 (61) | 12 | 95 (26) | 8 |
| **Rx medicines** | 103 (16) | 10 | 197 (47) | 17 | 286 (170) | 23 |

**Notes.**

Means are per capita, per year. OOP, out-of-pocket. Values are nationally representative except *n*'s.

No relationships were found between mean expenditures for IBD patients and race/ethnicity when comparing white to black patients, white to Hispanic patients, or white to non-white patients.

### Disproportionate spending on acute vs. non-acute care by insurance status and income in multivariate analyses

Multivariate logistic regressions paralleled annual per capita mean expenditure trends when examining the effect of insurance status on IBD expenditures. Figure 2B shows that for total expenditures, publicly insured patients had significantly higher odds of spending above the mean than privately insured (OR 2.13, CI [1.08–4.19]) and uninsured patients (OR 4.94, CI [1.26–19.47]). IBD patients with public insurance were more likely to spend more for acute care, defined as inpatient and emergency visits, compared to private or no insurance. Just as was seen with mean expenditures, the increased spending seen with public insurance was disproportionately due to high inpatient spending (public vs. private OR 2.82, CI [1.30–6.10]; public vs. uninsured OR 2.95, CI [1.02–8.54]). Emergency spending was also more likely to be above the mean with public insurance compared to private insurance (OR 2.50, CI [1.23–5.06]).

In contrast, privately insured IBD patients were more likely to spend more for non-acute care, defined as outpatient visits, office-based visits, and prescribed medicines. For

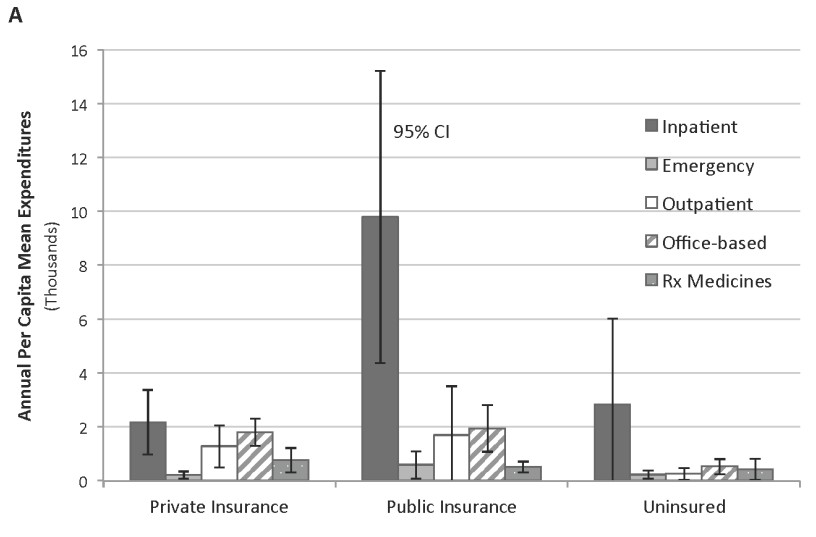

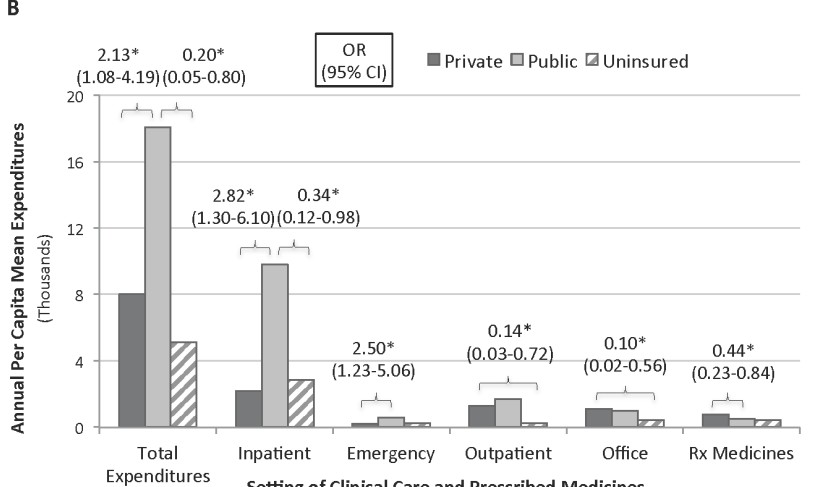

**Figure 2 Distribution of annual per capita/Odds of IBD expenditures.** (A) Distribution of annual per capita mean expenditures for IBD patients across various categories, by insurance status. Error bars are 95% confidence intervals. (B) Odds of IBD expenditures above the mean for the respective setting of clinical care, between two insurance status groups. Adjusted for age, sex, race/ethnicity, and comorbidities. $* p < 0.05$,

outpatient and office-based care, privately insured patients were significantly more likely to spend above the mean than the uninsured (outpatient OR 7.02, CI [1.39–35.40]; office OR 9.69, CI [1.78–52.67]), with no significant relationship to public insurance (Fig. 2B). For prescribed medicines, private insurance was more likely to spend above the mean than public insurance (OR 2.05, CI [1.08–3.88]).

Table 4 shows that poor IBD patients (<100% FPL) were less likely to spend more for non-acute care, compared to not poor IBD patients. Poor patients were significantly less likely to spend above the mean for outpatient (OR 0.38, CI [0.15–0.95]) and office-based care (OR 0.21, CI [0.07–0.62]). With a low $n = 41$ for poor patients, no other significant

**Table 4** Association between poverty and expenditures across clinical care settings.

| | Poor patients <100% FPL (*n* = 41) | |
| --- | --- | --- |
| | All expenditures (OR) (95% CI) | OOP expenditures (OR) (95% CI) |
| **Total** | 0.67 (0.31–1.48) | 0.91 (0.43–1.92) |
| **Acute care** | | |
| Inpatient | 1.01 (0.45–2.27) | 1.50 (0.58–3.92) |
| Emergency | 2.01 (0.95–4.22) | 1.93 (0.63–5.90) |
| **Non–acute care** | | |
| Outpatient | 0.38 (0.15–0.95)[*] | 0.40 (0.11–1.42) |
| Office–based | 0.21 (0.07–0.62)[*] | 0.38 (0.15–1.00) |
| **Rx medicines** | 0.56 (0.30–1.06) | 0.87 (0.39–1.92) |

**Notes.**

Odds of expenditures above the mean for the respective setting of clinical care for poor vs. not poor (*n* = 292) patients. Adjusted for age, sex, race/ethnicity, and comorbidities. FPL, federal poverty line; OOP, out-of-pocket. Values are nationally representative except *n*'s.

[*] $p < 0.05$.

differences were found between poor and not poor IBD patients for inpatient, emergency, prescribed medicine, or total expenditures.

No relationships were found in multivariate analyses comparing IBD patients' expenditures to race/ethnicity when comparing white to black patients, white to Hispanic patients, or white to non-white patients. No significant relationships were found regardless of whether poverty was included as a covariate or not.

## DISCUSSION

No known study to date correlates socioeconomic or racial/ethnic differences with health expenditures associated with different services and treatments in IBD. This level of expenditure detail is especially important in a disease such as IBD where a wide range of services and treatments and associated costs exist, potentially revealing patterns that total expenditure figures alone fail to capture (*Benchimol et al., 2011*; *Nguyen et al., 2010*; *Flasar et al., 2008*).

Using nationally representative data from 1996 to 2011, we determined that mean inpatient expenditures composed the highest proportion of IBD direct costs, above outpatient, emergency, office-based, and prescribed medicine costs. When IBD patients were stratified by insurance status, we found that publicly insured patients spent over double the mean expenditures of privately insured or uninsured patients, with differences of $10 K and $13 K, respectively. A combined analysis of expenditures by subcategories and insurance status revealed that inpatient costs are the overwhelming driver of public insurance's high expenditures (Table 3; Fig. 2A). In fact, after stratification by insurance status, privately insured and uninsured patients no longer showed disproportionately higher inpatient costs relative to the other subcategories, contrary to when all insurance groups were considered together in the IBD expenditure analyses of Table 2 and Fig. 1.

These mean expenditure data suggest that a primary driver of high IBD costs may be specifically localized to inpatient costs of one insurance group—public insurance. In

the current climate of healthcare reform and expenditure curtailment, especially among safety net programs such as Medicaid and SCHIP, our data reveal an intriguing source of potential economic waste and suggest a strategy for reducing the public burden of IBD healthcare costs. Further studies should explore the factors contributing to high inpatient utilization among publicly insured patients and evaluate means of reduction. One potential explanation is that publicly insured patients may reside in impoverished neighborhoods with less capacity to perform outpatient procedures, resulting in longer inpatient stays. *Nguyen et al. (2007)* first hypothesized this theory when they found that bowel resection rates decreased for those with Medicare, Medicaid, and the "self-paid".

Whether high inpatient costs are tied to inadequate outpatient and maintenance care, to unnecessary hospitalizations and overtreatment, or to yet unknown factors, curbing inpatient costs may have the additional benefit of improving health outcomes. Even beyond public insurers and insurees, a detailed understanding of forces driving inpatient utilization may help improve efficiency in IBD care for managed care organizations, hospitals, and their patients.

The uninsured were found to have significantly lower mean expenditures for outpatient care than publicly or privately insured patients. Similarly, poor IBD patients (<100% FPL) were less likely to spend above mean values for non-acute care in outpatient and office-based settings, when compared to not poor patients. This trend for the poor and underinsured to utilize less outpatient and office-based care was also seen in studies on the rates of CD-related bowel surgery, the use of laparascopic subtotal colectomy for UC, and access to urgent ambulatory care follow-up appointments (*Nguyen et al., 2007*; *Asplin et al., 2005*; *Medicaid Access Study Group, 1994*; *Greenstein et al., 2013*). The privately insured, on the other hand, were more likely to spend above mean values for non-acute care as well as prescribed medicines. We expected privately insured and not poor patients to spend more on non-acute care, perhaps due to a greater ability to pay OOP costs associated with these non-urgent visits. Greater non-acute care spending and less acute care spending may be associated with more desirable health outcomes as well, but those relationships remain to be studied.

Our findings also consistently reaffirm and expand previously published data. Our overall IBD expenditures and treated prevalence estimates approximate the current values in literature. Our annual per capita expenditures of $10,364 for CD and $7,827 for UC are within the range of previously published values of $8,265 and $11,129 for CD, and $5,066 and $7,706 for UC as published by Kappelman and Gunnarsson, respectively (*Gunnarsson et al., 2012*; *Kappelman et al., 2008*). Our treated prevalence values, despite missing IBD patients without medical events due to the nature of MEPS data collection, still approximate disease prevalences in literature (*Kappelman et al., 2007*; *Kappelman et al., 2013*; *Loftus, Schoenfeld & Sandborn, 2002*). This study's averaging of data over the years from 1996 to 2011 also affects the prevalence values, since prevalence rates have been on a steady rise (*Kappelman et al., 2013*; *Loftus, Schoenfeld & Sandborn, 2002*).

The strengths of the MEPS database lies in its in-depth, in-person survey design combined with insurer/employer and medical provider components allowing for an

unusually comprehensive single source of nationally representative information covering a broad range of clinical care with high granularity, prescription medicines, other medical conditions, socio-demographic information, and detailed insurer and OOP expenditure data. In comparison, the healthcare access and utilization literature for IBD has been restricted by the abundance of single-center or narrow-scope studies of clinical care-specific databases such as the Nationwide Inpatient Sample (NIS) (*Sewell & Velayos, 2013*). No prior study has analyzed IBD healthcare expenditures with respect to insurance status and socio-demographic factors in a nationally representative sample. A limitation of MEPS is the relatively small sample sizes once stratified by variables of interest. We therefore suspect that even more statistically significant and policy-relevant differences may exist that this study lacked enough power to demonstrate; for example, we may have missed a significant difference in inpatient expenditures between poor and not poor IBD patients.

In conclusion, this study presents comprehensive, nationally representative estimates of detailed expenditure data as they relate to disease type, insurance status, and poverty. These findings can inform IBD-related health policy, guide further analysis of inpatient utilization of publicly insured IBD patients as the main driver of IBD spending, and support IBD advocacy and economic research.

### Abbreviations

| | |
|---|---|
| **CD** | Crohn's disease |
| **FPL** | federal poverty line |
| **HRQL-CI** | health-related quality of life comorbidity index |
| **IBD** | inflammatory bowel disease |
| **MEPS** | Medical Expenditures Panel Survey |
| **OOP** | out-of-pocket |
| **OR** | odds ratio |
| **UC** | ulcerative colitis |

### Funding

Authors received support from the Stanford Medical Scholars Research Program of Stanford University School of Medicine. KTP is supported by NIH K08 DK094868. JB is supported by CDEHA P30 AG17253 and NIH R21 AG041112. The manuscript contents are solely the responsibility of the authors, performed independent of all funding sources, and do not necessarily represent the official views of the NIH or Stanford University. The funders had no role in study design, data collection and analysis, decision to publish, or preparation of the manuscript.

## Grant Disclosures

The following grant information was disclosed by the authors:
Stanford Medical Scholars Research Program of Stanford University School of Medicine.
NIH: K08 DK094868.
CDEHA: P30 AG17253.
NIH: R21 AG041112.

## Competing Interests

The authors declare there are no competing interests.

## Author Contributions

- Michelle D. Park conceived and designed the experiments, performed the experiments, analyzed the data, wrote the paper, prepared figures and/or tables.
- Jay Bhattacharya conceived and designed the experiments, reviewed drafts of the paper, technical expertise in MEPS database.
- KT Park conceived and designed the experiments, performed the experiments, analyzed the data, contributed reagents/materials/analysis tools, wrote the paper, reviewed drafts of the paper, wrote grant for Financial support NIH DK094868.

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
