# Peer review of "Differences in healthcare expenditures for inflammatory bowel disease by insurance status, income, and clinical care setting"

_PeerJ, doi:10.7717/peerj.587_

## Round 0.1 · original submission · Minor Revisions

Reviewers including myself are happy with the paper and are willing to accept it for publication pending some minor revisions. We are concerned about the study not including infliximab since this is the more common agent used. One reviewer suggested to include an economist for comments but besides the difficulty in getting one, others did not find the results too difficult to comprehend. Otherwise, the article is interesting and adds to the limited existing knowledge in this area.

Reviewer 1 ·

Basic reporting

See below

Experimental design

See below

Validity of the findings

See below

Additional comments

This is an interesting paper dealing with healthcare expenditures for inflammatory bowel disease (IBD). I have some comments to the present manuscript version.
1. The authors should notice that IBD does not fulfil the criteria for being an autoimmune disease, please check e.g., G, Behr MA, Lancet 2010; 376: 202.
2. In describing the increasing prevalence of IBD please use a more updated reference like e.g. Molodecky NA, Gastroenterology 2012; 142: 46

3. When comparing IBD with other diseases the authors should additionally include comparable inflammatory diseases like rheumatoid arthritis, multiple sclerosis and psoriasis (disorders affecting other organs than the intestine).
4. The authors should define all abbreviations in the manuscript, e.g. SES-strata, ED visits and various other ones not defined. Please, keep in mind that the audience of readers might be rather broad.
5. When mentioning immunomodulators it is suggested to write “….immunomodulators, that is thiopurines (6-mercaptopurins and azathioprine) and methotrexate”.
6. When describing TNF-inhibitors it seems that the author only included costs for adalimumab. However infliximab was the first drug marketed and later on certolizumab pegol and golimumab were added (e.g. Nielsen OH, N Engl J Med 2013; 369: 754). If Infliximab really has been deleted from the analyses this might seriously hamper the outcome of the estimate of healthcare costs, which might be much higher than stated in the present manuscript version?
7. Reference 28 might be used but the authors should check for newer and more updated references, especially as data published in a systematic review from 2002 represents data from the pre biologics era (and with higher surgery rates in the 80’s and 90’s than today). Thus, manuscripts like Lin KK, Am J Gastroenterol 2013; 108: 1824 and Wang YR, Digestion 2013; 88: 20 plus various others published in the meantime should be considered.
8. The reference list should be using the same style, e.g. the journal name should either be abbreviated or mentioned in full – but consistently! The authors need to read Instruction for Authors ones more.

·

Basic reporting

This longitudinal analysis aimed at describing IBD related expenditures based on insurance and household income in various settings in the United States. The study design and concept is novel and adds to the limited existing literature in IBD and health disparities.

The introduction does an adequate job to demonstrate how the work fits into the broader field of knowledge. The authors may want to include the following study:

Sewell JL1, Yee HF Jr, Inadomi JM. Inflamm Bowel Dis. 2010 Feb;16(2):204-7. doi: 10.1002/ibd.21008.

This study showed that the proportion of hospitalizations for IBD increased significantly among minority patients.

Experimental design

The authors used the Medical Expenditure Panel Survey (MEPS) is a nationally representative database. Under the subheading Study population and variables:
“IBD-related medications were identified using pharmacy-reported prescription names. We identified immunomodulators (6-mercaptopurine, azathioprine, and methotrexate), anti-TNF agents (adalimumab)…”

Was there a reason infliximab was not included?

Validity of the findings

The discussion adequately points out the strengths and weaknesses of the study.
As pointed out in the discussion, the sample size is relatively small and may lack the power to further analyze the data in detail.

Additional comments

This is a novel study that demonstrated a high inpatient utilization among publicly insured patients. Concerted, prospective multicenter efforts are needed to address the underlying causes for disparities driving these costs.

·

Basic reporting

Excellent article since no prior study has analyzed the IBD health care expenditure with respect to insurance status and social-demographic details in a well described nationally representative sample.
However, the sample size is relatively small as compared to the non-IBD population and it may possibly represent the most complex form of IBD cohort.
The potential factors that might contribute to the high inpatient cost for the publicly insured patients were not well informed. Were they related to the complexity and duration of the disease? And were involved in the IBD-associated malignancy that contribute more cost?

Experimental design

For the survey weights and weighting process including an adjustments for non-responses over time along with calibration to independent population warrant further explanation. What kind of standard calibration used against independent population?
There is no mention about the duration of disease for IBD patients which may indicate the chosen publicly insured patients who run more aggresive and malignat disease course.
For the IBD-related medicine, why adalimumab alone was calculated and not infliximab as it was the first biological agent used in the treatment of IBD.The author also mention prednisone and what about the usage of intravenous corticosteroids that might indicate hospitalization and hence more cost for IBD patients.
In my opinion, the IBD-related medicine such as laxative, anti-diarrhoea,PPI and H2 antagonist are not the IBD-related medicine and this can top-up the necessary cost in this analysis. PPI perhaps should be used in the case of upper GI IBD cases.
The other issues was the author did not describe the IBD-related medicine adverse effects that required hospitalization which can add more cost but not related to IBD itself.

Validity of the findings

Overall is a good findings.
The only issue is MEPS only collect information on condition associated with medical event and is this include IBD-related medicine which cause the significant adverse effect and necessitate hospitalization? And is there any adjustment require?

What are the general waiting list time for out-patient appointment for public patient in the USA? If is too long, that might explain the in-patient treatment seek for publicly insured patient

Additional comments

Overall, a very well done for this first time publication looking at the mentioned association between IBD health care cost with respect to insurance status and social-demographic details.
Congrats!

---

## Round 0.2 · accepted · Accept

The revised version has adequately addressed the concerns raised.